# The Psychological Impact of COVID-19 Admission on Families: Results from a Nationwide Sample in Greece

**DOI:** 10.3390/children9121933

**Published:** 2022-12-09

**Authors:** Despoina Gkentzi, Konstantinos Mhliordos, Ageliki Karatza, Xenophon Sinopidis, Dimitra Dimopoulou, Eirini Eleftheriou, Maria Tsolia, Artemis Mavridi, Eugenia Miliara, Vassiliki Papaevangelou, Eleni Vergadi, Emmanouil Galanakis, Gabriel Dimitriou, Sotirios Fouzas

**Affiliations:** 1Department of Pediatrics, Medical School, University of Patras, 26504 Rio Patras, Greece; 2Department of Pediatrics, Second Department of Paediatrics, School of Medicine, P. and A. Kyriakou Children’s Hospital, National and Kapodistrian University of Athens, 11527 Athens, Greece; 3Department of Pediatrics, Third Department of Paediatrics, School of Medicine, Attikon General Hospital, National and Kapodistrian University of Athens, 12462 Athens, Greece; 4Department of Paediatrics, School of Medicine, University of Crete, 71500 Heraklion, Greece

**Keywords:** COVD-19 admission, stress, parents

## Abstract

The aim of the present study was to assess the psychological impact of hospitalization during the COVID-19 pandemic on parents and their offspring. We performed a nationwide cross-sectional study in Greece based on an Internet questionnaire survey. A convenience sample of parents whose offspring had been hospitalized due to COVID-19 (including multisystem inflammatory syndrome in children, MIS-C), diagnosed with COVID-19 but not hospitalized, and hospitalized for another reason during the pandemic were enrolled. Parental stress was assessed using the Perceived Stress Scale (PSS) and the Revised Impact of Event Scale (IES-R) tools, and childhood mental wellbeing with the Children’s Revised Impact of Event 13 (CRIES-13) scale. Out of 214 received responses, stress levels were significantly higher in parents whose children had been admitted for COVID-19 or MIS-C versus those not admitted or admitted for other reasons (*p* < 0.001, for PSS/IES-R). Parental and childhood stress levels were correlated. In the multivariable linear regression analysis, children’s hospitalization because of COVID-19 or MIS-C, younger parental age, the existence of comorbidities, and another family member’s hospitalization because of COVID-19 were independent factors for higher stress. In light of the above, stricter hospital admission criteria for COVID-19 could be implemented, and psychological support for eventually admitted families may be beneficial.

## 1. Introduction

Coronavirus disease 2019 (COVID-19) has been identified as a pandemic with a great impact on the population’s wellbeing [1]. Historically, infectious disease outbreaks have had a severe effect on mental health [2,3]. During the COVID-19 pandemic, social and sanitary measures, including quarantine and the widespread use of masks, were implemented in order to prevent viral transmission [4,5,6]. The large spread of the disease, the high mortality rate, and the encircling uncertainty caused severe psychological distress and have had a negative effect on society’s emotional wellbeing [2,3]. Although the clinical symptoms of children infected with COVID-19 are milder than those of adults, and mortality is rare, studies showed that parents and children are especially vulnerable to feelings of emotional distress such as anger, depression, and irritability [2]. Many factors exacerbate their mental wellbeing, including quarantine, infection fear, frustration, and financial loss [5,6,7,8,9,10,11,12,13]. Many studies reported that teenagers were particularly affected, as quarantine caused a sudden interruption of all their social activities [6,7,14,15]. During lockdowns, they engaged in less physical activity, experienced irregular sleeping patterns, and followed unhealthy eating habits. All these lead to increased symptoms of anxiety, irritability, insomnia, and depression [5,6,14,15].

During the COVID-19 pandemic, parents also showed increasing symptoms of anxiety, fatigue, sleeping disorders, and depression [2,10]. Reports also indicated that parents experiencing social isolation, unemployment, and financial instability showed increased signs of child neglect and maltreatment [8,9]. Parents of hospitalized children during the pandemic showed higher symptoms of stress and anxiety than those of parents of children who were not hospitalized [12,13]. Furthermore, parents of children with comorbidities such as immunosuppression and cystic fibrosis showed symptoms of emotional distress [3]. Despite the fact that there are many reports about the impact of the pandemic on parents and children’s mental wellbeing, data on parental stress level during their children’s hospitalization with COVID-19 are scarce [12,13,14,15,16,17]. The aim of this present study was to the assess stress levels due to hospitalization during the COVID-19 pandemic on parents and their offspring. Our main hypothesis is that parents whose COVID-19-positive children were hospitalized experienced higher stress levels than those who had been cared for on an outpatient basis or had been admitted for a reason other than COVID-19.

## 2. Materials and Methods

### Study Participants

This was a nationwide cross-sectional study in Greece based on an Internet questionnaire survey (e-survey). The study adhered to the CHERRIES guidelines of the EQUATOR network (Appendix A). Participants were identified from the patients’ lists of four tertiary paediatric departments in Greece with dedicated COVID-19 wards (University General Hospital of Patras, University General Hospital of Herakleion, Athens Children’s Hospital of P. and A. Kyriakou, and Attikon General Hospital) and were contacted via email or a personal message on social media platforms. A convenience sample of parents whose offspring had been hospitalized due to COVID-19 (including multisystem inflammatory syndrome in children, MIS-C), diagnosed with COVID-19 but not hospitalized, and hospitalized for another reason during the pandemic were enrolled. Potential participants were identified from the lists of inpatients and outpatients in the participating hospitals. Parents were initially contacted over the phone, and asked for their email address and subsequently to send the questionnaire via email. A total of 250 invitations were eventually sent. More details on the recruitment process are given in the Appendix A.

## 3. Questionnaire

The questionnaire was developed in the Google Forms platform. It consisted of four sections: (1) introduction; (2) section on general information and demographics; (3) section on children’s medical history; (4) section on psychometric testing, which included the validated Greek versions of the Perceived Stress Scale (PSS) (10 questions) [18,19], the Revised Impact of Event Scale (IES-R) (22 questions) [20,21,22], and the Children’s Revised Impact of Event 13 (CRIES 13) scale [23,24,25]. PSS is a widely used test for measuring stress perception; it captures the feelings and thoughts of participants, and identifies how unpredictable, uncontrollable, and overloaded they find their life [18,19]. IES-R is a self-report tool that measures the subjective psychological response caused by a stressful event [20,21,22]. Children’s psychological distress was evaluated using CRIES 13 [23,24,25], which consisted of 13 questions answered by the parent. Potential participants were initially directed to the introduction page, where information regarding the study aims and a consent form were available. The parents that did not consent were directed to the end of the survey. The total number of questions ranged from 44 to 56, depending on the study group. Answering all questions was mandatory, while the participants could check their answers or exit the survey at any time. The questionnaire was available for answering from 1 June 2021 (12:00 a.m.) up until 31 March 2022 (11:59 p.m.), and the responses were stored in a secure database.

The study was approved by the Ethics Committee of the University General Hospital of Patras (decision number 292/08-06-2021).

## 4. Statistical Analysis

Continuous variables, including PSS, IES-R and CRIES 13 scores, are presented as mean ± standard deviation (SD) and range. The Spearman factor was used to expose the correlations among the PSS, IES-R, and CRIES 13 scores, while comparisons among the multiple study groups were performed with the Kruskal–Wallis and Dunn’s post hoc tests. An IES-R score equal to 38 was considered diagnostic for post-traumatic stress disorder [21]. Univariable and multivariable linear regression models were used to assess the multiple-factor effect on PSS and IES-R scores. Only factors that were statistically significantly associated with PSS and IES-Rin in the univariable regression model were included in the multivariable analysis. Statistical analyses were performed with IBM SPSS version 27 (IBM Corp., Armonk, NY, USA).

## 5. Results

A total of 250 invitations were sent via email. The number of enrolled visitors that entered the introduction page was 214. All of them eventually completed the survey (214/250, participation rate 85.6%). One parent from each family completed the survey. Of the study participants, 56,1% lived in the region of Western Greece, 31.3% in Athens, and 12.6% in Crete. Of the parents who completed the questionnaire, 58 had children that were hospitalized due to COVID-19, 11 due to MIS-C, and 41 for another reason during the pandemic, whereas 104 were diagnosed with COVID-19 but not hospitalized.

The demographics of the 214 participants are presented on Table 1, and Table 2 presents the comorbidities of 49 children as reported by their parents (22.9%).

PSS and IES-R scores were correlated (rho = 0.713, *p* < 0.001) (Figure 1). There was also a strong correlation between CRIES-13 (16.9 ± 15.8) and PSS (rho = 0.547, *p* < 0.001), and between CRIES-13 and IES-R (rho = 0.619, *p* < 0.001) (Figure 2). 

The PSS for parents whose offspring had been hospitalized due to MIS-C (mean 27.3 ± 5.9, median 26, range 18–36) was significantly higher than those whose children had been infected with COVID-19 but not hospitalized (16.8 ± 6.9, median 20, range 1–32, *p* < 0.001), and than those that had children who had been hospitalized for other reasons during the same period (17.8 ± 6.2, median 18, range 6–33, *p* < 0.001) (Figure 3). Similarly, the PSS of parents whose children had been hospitalized due to COVID-19 (mean 22 ± 5,6, median 23, range 13–34) was significantly higher than that of parents whose children had not been hospitalized (*p* < 0.001), and that of parents whose children had been hospitalized for other reasons (*p* < 0.001) (Figure 3).

The IES-R of parents whose children were hospitalized due to MIS-C (mean 45.5 ± 10.8, median 40, range 38–63) was significantly higher than that of parents whose children had been infected with COVID-19 but not hospitalized (27.2 ± 19, median 24, range 0–68, *p* = 0.021) and that of parents whose children had been hospitalized for other reasons during the same period (26.4 ± 15.7, median 26, range 4–56, *p* < 0.001) (Figure 4). Similarly, the IES-R of parents whose offspring had been hospitalized because of COVID-19 (mean 38.4 ± 10, median 35, range 28–74) was significantly higher than that of parents whose children had had COVID-19 but had not been hospitalized (*p* = 0.032), and that of parents whose offspring had been hospitalized for other reasons (*p* < 0.001) (Figure 4).

An IES-R score higher than 38 (diagnostic for post-traumatic distress) (19) was noted in 72.7% (N = 8) of the parents whose children had been hospitalized because of MIS-C, in 58.6% (N = 34) of those whose children had been hospitalized because of COVID-19, in 41.3% (Ν = 43) of those with children that had had COVID-19 and had not been hospitalized, and in 31.7% (Ν = 13) of those that had been admitted to the hospital for other reasons (chi-square *p* = 0.011) (Figure 4). 

The multivariable linear regression analysis showed that children’s hospitalization because of COVID-19 or MIS-C, parental age, existence of comorbidities and another family member’s hospitalization because of COVID-19 were important and independent factors that had caused higher levels of emotional distress among parents (Table 3). 

Children’s stress levels with COVID-19 that had not been hospitalized were lower than those that had been hospitalized because of MIS-C (*p* < 0.001) or COVID-19 (*p* < 0.001) and than those of children that had been hospitalized for other reasons (*p* = 0.017). No significant statistical difference was noted among the last three groups (Figure 5), indicating that the hospitalization itself increases the stress level rather than the COVID-19 infection.

## 6. Discussion

In this study, conducted on a nationwide sample in Greece, we found that admission due to COVID-19 or MIS-C was associated with higher stress levels for both parents and children compared to outpatient COVID-19 management or admission for other reasons during the pandemic. Similar studies in the field are lacking. Few studies measured the psychological impact in adult inpatients using standardized psychometric tools, and concluded that admission for COVID-19 has a negative effect on mental health [26,27,28,29,30]. However, no comparison groups were included. In an Italian study recruiting parents of children admitted to a pediatric intensive care, unit ratings of psychological distress were compared between caregivers with no or severely restricted access and with limited access to PICUs [17]. Restrictions imposed on visitation policies in PICU during the pandemic negatively impacted families’ psychological wellbeing, and that could be explained well by the separation anxiety. Our study differs from the above, as the participating parents in all groups were accompanying their children throughout their admission. Another study from China [12] in the beginning of the pandemic showed that, compared to the prepandemic period, parents of hospitalized children experience higher stress levels. The latter was somewhat expected, as the pandemic has dramatically changed the hospital setting and function, and has increased the overall uncertainty and anxiety for the future. In our study, however, we assessed the stress levels with groups within the same period so that we could extrapolate findings about the actual COVID-19 admission effect irrespective of the background stress caused by the pandemic itself. For that reason, one of our comparison groups was the group of children admitted for other reasons. Hence, our results focus more on the actual admission for COVID-19. In addition, in a larger US study [13], a significant rise in mental health consultations for inpatient children during the pandemic was found, highlighting the overall and undisputable impact of the pandemic in the youth. However, no comparisons were performed regarding this impact among COVID-19 admissions versus admissions for other reasons or no admission following COVID-19 diagnosis. In the multivariate analysis, we also demonstrated that parental age, the existence of comorbidities, and another family member’s hospitalization because of COVID-19 were important and independent factors that caused higher levels of stress. Having a child with an underlying medical condition understandably increases parental concerns and was previously described in the setting of the COVID-19 pandemic [3]. Equally, having another family member admitted due to COVID-19 has a negative impact on the overall disease perception and, when it comes to children`s admission, may lead to higher stress. Lastly, younger parents showed higher scores in both PSS and IES-R scales, which could be partially explained by an insufficient prior understanding of the disease consequences and lower vaccination uptake levels. This could, in turn, lead to a higher psychological burden when their child eventually contracts COVID-19 and ends up admitted to the hospital.

In our cohort, admissions for MIS-C and COVID-19 had a similar psychological impact on parents and their offspring. One would expect a higher impact of admission for MIS-C versus COVID19 given the widespread publicly available information on the complications and severity of MIS-C versus the overall mild disease nature of COVID-19 in childhood. Our findings could be partially explained by the small number of patients in the MIS-C group or the phase of the pandemic during which they were actually recruited, which might have influenced the degree of available information on the complications and overall course of each clinical entity. Moreover, the strict infection control measures applied to COVID19 inpatients versus MIS-C PCR-negative patients that resulted in less frequent patient–staff interaction and hence higher insecurity levels may have accounted for the similar findings in these two groups.

Our study adds to the limited body of the literature in the field and raises further public health considerations. Given the increased knowledge that we currently have about the mild nature of COVID-19 in children and the psychological impact of an admission for COVID-19 for families, stricter inpatient admission criteria could perhaps be set to prevent a potentially unnecessary hospital stay and further consequences associated with that, such as increased cost, healthcare-associated infections, and staff workload. If the admission is deemed necessary, hospital protocols may also include the provision of psychological support for parents and children admitted for COVID-19, especially in prolonged hospital stays.

To our knowledge, this is the first study reporting on the psychological impact of admission for COVID-19 on the mental wellbeing of parents and their offspring. As an pragmatic study, it has several limitations that we have to acknowledge. First, the number of study participants was small. However, these results could be country-representative, as the setting of COVID-19 wards is similar within the country. We cannot, however, generalize these findings to other countries, as cultural differences and the provision of mental health support vary between countries and may play a role in the observed psychological impact. In addition, our participants were asked to answer questions about an event that had occurred weeks or months ago, which might have underestimated their true stress level at the time of the admission (recall bias). Moreover, we did not compare the different phases of the pandemic, which might have shown a trend towards less stress, as our knowledge about the generally mild nature of COVID-19 in children has gradually increased. Regarding children’s stress, the CRIES-13 questionnaire provides an indirect assessment of childhood stress, as it is reported by their parents. Nonetheless, all three tools used in this study have been extensively used and validated in other studies in the field. Lastly, in this study, we used validated tools to assess stress as a marker of the psychological impact, but did not assess other parameters such as anxiety or depression. This could serve as the goal of a future study in the field to enable a better understanding of the psychological impact of COVID-19 admission in families. In addition, parents with diagnosed mental health problems may have deliberately refused to participate in the study.

## 7. Conclusions and Recommendations

In conclusion, in the present study, we demonstrated that inpatient admission for COVID-19 is a significant predictor of stress in both parents and their children. Parental and offspring’s stress levels, as those reported by their parents, were correlated. Given the overall body of knowledge in the literature that we have about COVID-19 in children, the unpredicted course of the pandemic in the forthcoming years, and the psychological impact of the admission for families, stricter inpatient admission criteria could be implemented. In addition, psychological support for parents and families that have been admitted for COVID-19 may be warranted if we also bear in mind the overall underlying burden of the pandemic on mental health.

## Figures and Tables

**Figure 1 children-09-01933-f001:**
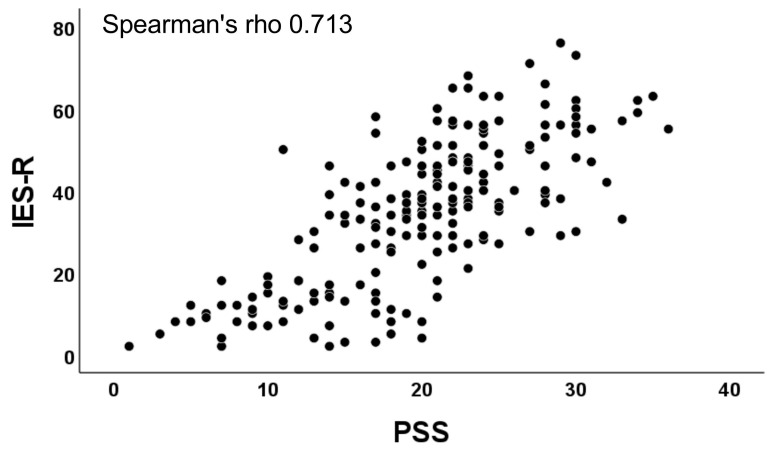
Correlation between PSS and IES-R scores.

**Figure 2 children-09-01933-f002:**
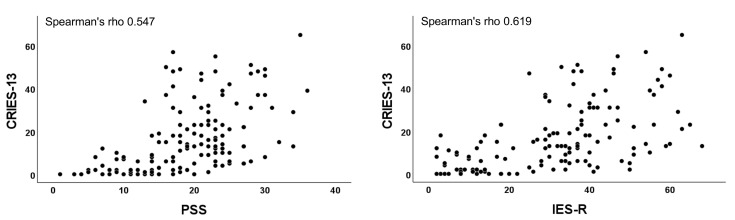
Correlation between CRIES-13 and PSS, and CRIES-13 and IES-R scores.

**Figure 3 children-09-01933-f003:**
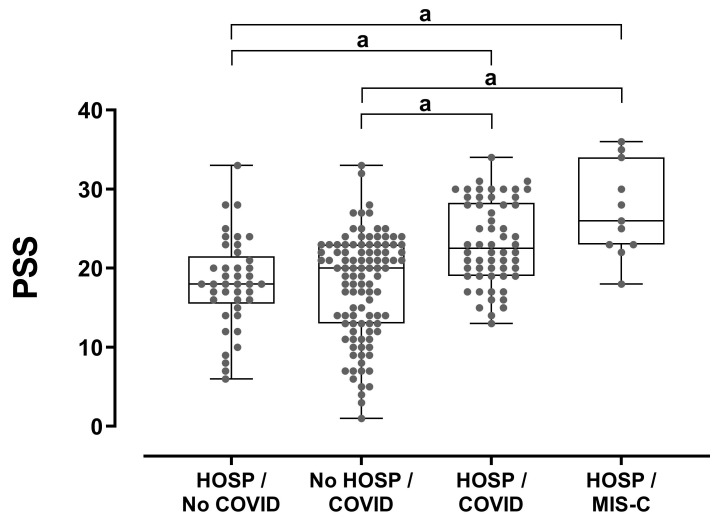
Parental stress levels in relation to hospitalization and COVID-19 status. Between-group comparisons were performed using Kruskal–Wallis and Dunn’s post hoc tests. HOSP: hospitalization. a: represents the comparison groups.

**Figure 4 children-09-01933-f004:**
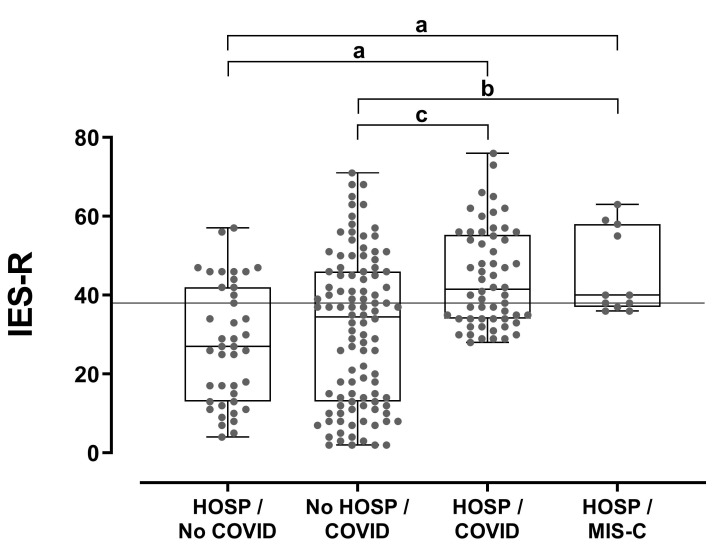
Parental post-traumatic stress levels in relation to hospitalization and COVID-19 status. Between-group comparisons were performed using Kruskal–Wallis and Dunn’s post hoc tests. a: *p* < 0.001, b: *p* = 0.021, c: *p* = 0.032. The horizontal line (IES-R = 38) denotes the diagnostic cut-off for post-traumatic disease. HOSP: hospitalization.

**Figure 5 children-09-01933-f005:**
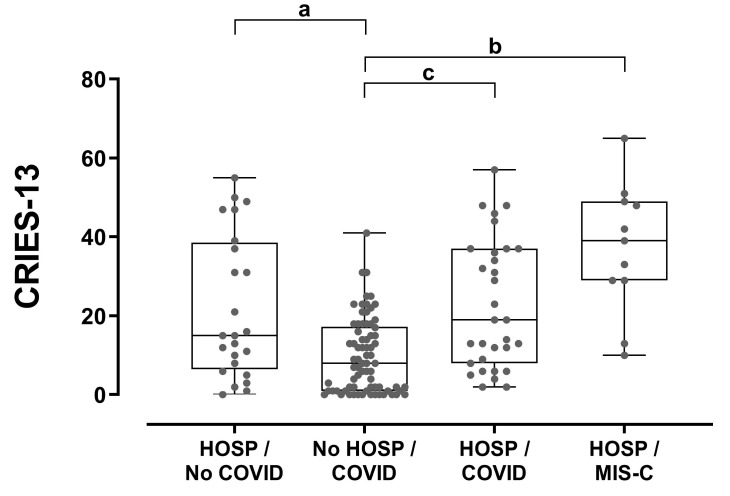
Children’s stress levels (as reported by the parents) in relation to hospitalization and COVID-19 status. Between-group comparisons were performed using Kruskal–Wallis and Dunn’s post hoc tests. a: *p* = 0.017, b: *p* < 0.001, c: *p* = 0.001. HOSP: hospitalization.

**Table 1 children-09-01933-t001:** Participants’ demographics.

Total number of participants	214
Mothers, n (%)	146 (68.2)
Parent’s age (years) *	35.4 ± 9.9 (18–54)
Child’s age, (years) *	7.1 ± 5.4 (0.1–18)
Greek nationality, n (%)	196 (91.6)
Education, n (%)	
Primary	7 (3.3)
Secondary, junior high school	24 (11.2)
Secondary, senior high school	66 (30.8)
Tertiary—no university	42 (19.6)
Tertiary—university	75 (35)
Residence, n (%)	
Urban	159 (74.3)
Semiurban	42 (19.6)
Rural	13 (6.1)

* mean ± SD (min—max).

**Table 2 children-09-01933-t002:** Children’s comorbidities.

N	49
Diseases	
Cardiovascular	5
Respiratory	5
Diabetes mellitus	12
Immunodeficiencies	4
Renal	2
Neurological	8
Allergies	7
Gastroenterological, anemia, etc.	6

**Table 3 children-09-01933-t003:** Multivariable linear regression on factors that affect PSS and IES-R scales.

	PSS	IES-R
Hospitalization because of COVID-19/MIS-C	0.436	0.510
Parent’s age	−0.345	−0.175
Comorbidity	0.215	0.212
Other family member’s hospitalization	0.338	0.251

Multivariable linear regression models were used. The factors that were not important and were excluded from the models were the following: child’s age, residence, parents’ education, another family member’s COVID-19 infection, number of children in the family. All factors were important on a *p* < 0.001 level. PSS: perceived stress scale; IES-R: impact of event scale revised.

## Data Availability

Additional study data are available on request.

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
