# Peer review of "The Psychological Impact of COVID-19 Admission on Families: Results from a Nationwide Sample in Greece"

_children, 2022, doi:10.3390/children9121933_

Round 1

Reviewer 1 Report

This article deals with a very important topic, the psychological impact of hospitalization during the COVID-19 pandemic on parents and their offspring. The importance and relevance are beyond doubt. Ethical considerations are also taken into account. However, the manuscript would benefit by addressing the following.

Introduction:

The authors indicated that “stress and anxiety were also highly correlated with the risk of infection with COVID-19 in the hospital”. It would be necessary to clear this result in the introduction because the causality is not clear.

Methods:

-        About instruments: indicate the data validation, the specific punctuation of each instrument and the range of diagnosis.

-        Sample: The number of participants not is totally clear in the text. It would be necessary to make a flowchart indicating the total number of parents, children and group of illness. It is not clear if both parents did the survey or if was only one.

Results:

-        The participation percentage has to be about the total of 250 invitations, not about the 214 visitors participated. So, it wouldn’t be 100%.

-        Figure 1 is not explained in the text.

-        First paragraph of the seven page doesn't indicate the table or figure that makes reference.

-        Regarding the multivariable linear regression model, it would be necessary to explain why the variables that were excluded were not considered important.

Discussion:

The authors indicated that “In our study though we have assessed anxiety levels with groups within the same period so that we can extrapolate findings about the actual COVID-19 admission effect irrespective of the background anxiety caused by the pandemic itself”. However, the anxiety levels were not evaluated in this manuscript. And the same occur in the conclusions part.

Finally, it would be interesting to provide a solid justification in the limitation section for why a measure of anxiety and depression was left out of this investigation. In the literature now available, stress is far too frequently linked to depressive and anxious symptoms. Likewise, only the effect of stress has been measured despite the title of the publication referring to "psychological impact."

Minor comment: Unify the way references are placed, before or after a dot or comma.

Author Response

Response to Reviewer`s  1 comments

This article deals with a very important topic, the psychological impact of hospitalization during the COVID-19 pandemic on parents and their offspring. The importance and relevance are beyond doubt. Ethical considerations are also taken into account. However, the manuscript would benefit by addressing the following.

We would like to thank the reviewer for the positive feedback and the opportunity to improve our manuscript with the useful comments

Introduction

The authors indicated that “stress and anxiety were also highly correlated with the risk of infection with COVID-19 in the hospital”. It would be necessary to clear this result in the introduction because the causality is not clear.

Methods

- About instruments: indicate the data validation, the specific punctuation of each instrument and the range of diagnosis.

The translation in Greek of the instruments we have used was validated in previous studies (references 18-25).

 As commented by the reviewer only stress levels were measured to assess the psychological impact.

As noted in the methods section of the manuscript an IES-R score equal to 38 was considered diagnostic for post-traumatic stress disorder

- Sample: The number of participants not is totally clear in the text. It would be necessary to make a flowchart indicating the total number of parents, children and group of illness. It is not clear if both parents did the survey or if only one was.

Thank you for the comment. To make the number of participants clearer we have added the following in the result section: A total of 250 invitations were sent via email. The number of enrolled visitors that entered the introduction page was 214. All of them eventually completed the survey (214/250, participation rate 85,6%). One parent from each family completed the survey.

Results:

- The participation percentage has to be about the total of 250 invitations, not about the

214 visitors participated. So, it wouldn’t be 100%.

Thank you for this comment. We have amended the response rate calculated as 214/250 hence 85.6%.

A total of 250 invitations were sent via email. The number of enrolled visitors that  entered the introduction page was 214. All of them eventually completed the survey (214/250, participation rate 85,6%). One parent from each family completed the survey.

- Figure 1 is not explained in the text.

We have now added the following to explain the Figure 1 in the text: PSS and IES-R scores were correlated (rho = 0,713, P<0,001)(Figure 1). 

- First paragraph of the seven page doesn't indicate the table or figure that makes reference.

We have now added Figure 4 at the end of this paragraph as this is where all the information is shown.

- Regarding the multivariable linear regression model, it would be necessary to explain why the variables that were excluded were not considered important.

The variables that were not included in the multivariable liner regression model were those factors that were found not statistically significantly in the univariate analysis.

To clarify this, we have added the following in the statistical analysis section: Univariable and multivariable linear regression models were used to assess multiple factors effect on PSS and IES-R scores. Only factors that were found to be statistically significantly associated with PSS and IES-R in the univariable regression model were included in the multivariable analysis

Discussion:

The authors indicated that “In our study though we have assessed anxiety levels with groups within the same period so that we can extrapolate findings about the actual COVID-19 admission effect irrespective of the background anxiety caused by the pandemic itself”. However, the anxiety levels were not evaluated in this manuscript. And the same occur in the conclusions part.

Finally, it would be interesting to provide a solid justification in the limitation section for why a measure of anxiety and depression was left out of this investigation. In the literature now available, stress is far too frequently linked to depressive and anxious symptoms.

Likewise, only the effect of stress has been measured despite the title of the publication referring to "psychological impact."

We would like to thank the reviewer for the very useful comment. Following this we have amended the manuscript from the introduction to conclusion specifying that psychological impact was assessed via stress levels only and not anxiety or depression. All the changes are highlighted in the revised manuscript version.

We have also acknowledged this issue in the limitation session following your useful comment as below: Finally, in this study we have used validated tools to assess stress as a marker of the psychological impact and did assess other parameters such as anxiety or depression. It is also worth mentioning at this point that parents with diagnosed mental health problems may have deliberately refused to participate in the study.

Minor comment: Unify the way references are placed, before or after a dot or comma.

We have now placed all the references after the dot

Reviewer 2 Report

The paper : " The psychological impact of COVID-19 admission on families: results from a nationwide sample in Greece " is an interesting and important article. The aim of this research was to assess the psychological impact of hospitalization during the COVID-19 pandemic on parents and their offspring.

Participants of the study were identified from the patients’ lists of four Tertiary Paediatric Departments in Greece with dedicated COVID-19 wards. A total of 250 invitations were sent via email. The number of enrolled visitors that eventually entered the introduction page was 214. I don't understand why such a number can be a nationwide study.

The reliability, validity and development of the study tools were not specified.

The multivariable linear regression analysis showed that child’s hospitalization because of COVID-19 or MIS-C, parental age, existence of comorbidities and another family member’s hospitalization because of COVID-19 were important and independent factors that caused higher levels of emotional distress among parents. However, the regression results are not described in detail, such as the explainable variation? The results of parental age, existence of comorbidities and another family member’s hospitalization because of COVID-19 were important and independent factors that caused higher levels of emotional distress among parents also have not been explained and discussed.

Author Response

Response to Reviewer`s 2 comments

The paper: " The psychological impact of COVID-19 admission on families: results from a nationwide sample in Greece " is an interesting and important article. The aim of this research was to assess the psychological impact of hospitalization during the COVID-19 pandemic on parents and their offspring. Participants of the study were identified from the patients’ lists of four Tertiary Paediatric Departments in Greece with dedicated COVID-19 wards. A total of 250 invitations were sent via email. The number of enrolled visitors that eventually entered the introduction page was 214. I don't understand why such a number can be a nationwide study.

Τhank you for positive comments on our manuscript and the opportunity to revise it. Not all parents were willing to participate in the study and therefore from all the phone calls made to ask permission to get the email address and send the email 250 parents agreed to give us their email address for research purposes.We have clarified this in the revised version of the article to avoid reader confusionas follows: Potential participants were identified from the lists of inpatients and outpatients in the participating hospitals. Parent were initially contacted over the phone and asked permission to get their email address and subsequently send the questionnaire via email. A total of 250 invitations were eventually sent

The reliability, validity and development of the study tools were not specified.

Thank you for the comment. For the purpose of the study we have used already used validated tools. The translation in Greek of the instruments we have used was performed and validated in previous studies listed in the references 18 up to 2.

The multivariable linear regression analysis showed that child’s hospitalization because of COVID-19 or MIS-C, parental age, existence of comorbidities and another family member’s hospitalization because of COVID-19 were important and independent factors that caused higher levels of emotional distress among parents. However, the regression results are not described in detail, such as the explainable variation? The results of parental age, existence of comorbidities and another family member’s hospitalization because of COVID-19 were important and independent factors that caused higher levels of emotional distress among parents also have not been explained and discussed.

Following your comment we have added the following in the discussion session: In the multivariate analysis we have also demonstrated that parental age, existence of comorbidities and another family member’s hospitalization because of COVID-19 were important and independent factors that caused higher levels of stress. Having a child with an underlying medical condition will understandably increase parental concerns and has been previously described in the setting of the COVID-19 pandemic [3]. Equally, having another family member admitted due to COVID-19 has a negative impact on the overall disease perception and when it comes to children`s admission may lead to higher stress. Finally, younger parents showed higher scores in both PSS and IES-R scales which could be partially explained by insufficient prior understanding of the disease consequences as well as lower vaccination uptake levels. This could in turn lead to higher psychological burden when their child eventually contracts COVID-19 and ends up admitted to the hospital.

Round 2

Reviewer 1 Report

Although the authors have completed most of the changes, they have not clarified the following:

Introduction

The authors indicated that “stress and anxiety were also highly correlated with the risk of infection with COVID-19 in the hospital”. It would be necessary to clear this result in the introduction because the causality is not clear. They just changed the sentense without giving any explanation.

Discussion:

it would be interesting to provide a solid justification in the limitation section for why a measure of anxiety and depression was left out of this investigation. In the literature now available, stress is far too frequently linked to depressive and anxious symptoms.

They added "finally, in this study we have used validated tools to assess stress as a marker of the psychological impact and did assess other parameters such as anxiety or depression". But they didn’t measure anxiety or depression.

Author Response

Although the authors have completed most of the changes, they have not clarified the following:Introduction: The authors indicated that “stress and anxiety were also highly correlated with the risk of infection with COVID-19 in the hospital”. It would be necessary to clear this result in the introduction because the causality is not clear. They just changed the sentence without giving any explanation.

Following your comment  and to  avoid any reader confusion we have deleted this sentence  

Discussion: it would be interesting to provide a solid justification in the limitation section for why a measure of anxiety and depression was left out of this investigation. In the literature now available, stress is far too frequently linked to depressive and anxious symptoms. They added "finally, in this study we have used validated tools to assess stress as a marker of the psychological impact and did assess other parameters such as anxiety or depression". But they didn’t measure anxiety or depression.

Unfortunately we have not assessed anxiety and depression in this study. This could be targeted in a future study to assess more extensively the psychological impact of COVID-19 admission.

We have added the following in the limitations section: Finally, in this study we have used validated tools to assess stress as a marker of the psychological impact but did not assess other parameters such as anxiety or depression. The latter could serve as a goal of a future study in the field to enable a better understanding of the psychological impact of COVID-19 admission in families.

Reviewer 2 Report

Although some revisions have been made to the paper, I find several key points have not been adequately addressed. I don't understand why 214 participants can qualify as a nationwide study?

The multivariable linear regression analysis results are still not described in detail, such as the explainable variation. The results of parental age, existence of comorbidities were important and independent factors that caused higher levels of emotional distress among parents also have not been explained and discussed.

This conclusion of the present study demonstrated that inpatient admission for COVID-19 is a significant predictor of stress and anxiety for both parents and their children. Parental and offspring’s stress levels were correlated. However, there is not a measure of anxiety for both parents and children, nor stress for children in this study.

Author Response

Although some revisions have been made to the paper, I find several key points have not been adequately addressed. I don't understand why 214 participants can qualify as a nationwide study?

This is the number of parents that responded to the email invitation. Potential participants were identified from the lists of inpatients and outpatients in the participating hospitals. Parents were initially contacted over the phone and asked permission to get their email address and subsequently send the questionnaire via email. A total of 250 invitations were eventually sent. We understand that this number may sound small but low response rate in such studies is a well known limitation. As the participants were from all over the country we used the word nationwide.

We have acknowledged the small number of study participants as a study limitation as follows: First of all, the number of study participants was small. Yet these results could be country representative as the setting of COVID-19 wards in similar within the country. We cannot however generalize these findings to other countries as cultural differences and provision of mental health support vary between countries and may have a role in the observed psychological impact

The multivariable linear regression analysis results are still not described in detail, such as the explainable variation. The results of parental age, existence of comorbidities were important and independent factors that caused higher levels of emotional distress among parents also have not been explained and discussed.

Our results show that parental age, existence of comorbidities were important and independent factors that caused higher levels of emotional distress among parents. Following the reviewer`s comment we have added the following extra paragraph in the discussion section to provide possible explanation for these findings:

In the multivariate analysis we have also demonstrated that parental age, existence of comorbidities and another family member’s hospitalization because of COVID-19 were important and independent factors that caused higher levels of stress. Having a child with an underlying medical condition will understandably increase parental concerns and has been previously described in the setting of the COVID-19 pandemic [3]. Equally, having another family member admitted due to COVID-19 has a negative impact on the overall disease perception and when it comes to children`s admission may lead to higher stress. Finally, younger parents showed higher scores in both PSS and IES-R scales which could be partially explained by insufficient prior understanding of the disease consequences as well as lower vaccination uptake levels. This could in turn lead to higher psychological burden when their child eventually contracts COVID-19 and ends up admitted to the hospital.

This conclusion of the present study demonstrated that inpatient admission for COVID-19 is a significant predictor of stress and anxiety for both parents and their children. Parental and offspring’s stress levels were correlated. However, there is not a measure of anxiety for both parents and children, nor stress for children in this study.

We have deleted the word anxiety as this was not measured in our study (only stress was measured). With regards to stress levels of children this was indirectly assessed by parent using the CRIES tool. We have acknowledged this as one of our study limitations.

We have rephrased the sentence in the conclusion following your advice: Parental and offspring’s stress levels, as those reported by their parents, were correlated.